# The Effectiveness of Fluoroscopy-Guided Manual Lymph Drainage as Part of Decongestive Lymphatic Therapy on the Superficial Lymphatic Architecture in Patients with Breast Cancer-Related Lymphoedema: A Randomised Controlled Trial

**DOI:** 10.3390/cancers15051545

**Published:** 2023-02-28

**Authors:** Nele Devoogdt, Sarah Thomis, An De Groef, An-Kathleen Heroes, Ines Nevelsteen, Nick Gebruers, Wiebren A. A. Tjalma, Jean-Paul Belgrado, Chris Monten, Marianne Hanssens, Tessa De Vrieze

**Affiliations:** 1Department of Rehabilitation Sciences, KU Leuven—University of Leuven, 3000 Leuven, Belgium; 2Department of Vascular Surgery, Centre for Lymphedema, UZ Leuven—University Hospitals Leuven, 3000 Leuven, Belgium; 3Department of Cardiovascular Sciences, KU Leuven—University of Leuven, 3000 Leuven, Belgium; 4Department of Rehabilitation Sciences and Physiotherapy, University of Antwerp, MOVANT, 2610 Antwerp, Belgium; 5Multidisciplinary Breast Centre, UZ Leuven—University Hospitals Leuven, 3000 Leuven, Belgium; 6Multidisciplinary Breast Clinic and Multidisciplinary Oedema Clinic, Antwerp University Hospital, 2650 Antwerp, Belgium; 7Department of Medicine, University of Antwerp, MIPRO, 2610 Antwerp, Belgium; 8Lymphology Research Unit, Université libre de Bruxelles, 1070 Brussels, Belgium; 9Department of Radiotherapy, Ghent University Hospital, 9000 Ghent, Belgium; 10Department of Oncology, Centre for Oncology, General Hospital Groeninge, 8500 Kortrijk, Belgium

**Keywords:** breast neoplasm, lymphoedema, manual lymph drainage, lymphatic system, near infrared fluorescence lymphatic imaging, molecular imaging, indocyanine green

## Abstract

**Simple Summary:**

This multicentre randomised controlled trial investigated the added value of fluoroscopy-guided manual lymph drainage (MLD) (compared to traditional or placebo MLD) as part of the decongestive lymphatic therapy (DLT) for superficial lymphatic architecture in patients with chronic mild to moderate breast cancer-related lymphoedema (BCRL). No differences between the three groups were found regarding the change in the number of efferent superficial lymphatic vessels leaving the dermal backflow region in the total dermal backflow score, and in the number of superficial lymph nodes. As a consequence, this trial was not able to demonstrate the beneficial value of MLD for the other components of DLT on superficial lymphatic architecture in patients with chronic mild to moderate BCRL.

**Abstract:**

The objective of this trial was to investigate the effectiveness of fluoroscopy-guided manual lymph drainage (MLD), as part of decongestive lymphatic therapy (DLT), on the superficial lymphatic architecture in patients with chronic mild to moderate breast cancer-related lymphoedema (BCRL). This trial was a multicentre, double-blind, randomised controlled trial involving 194 participants with BCRL. Participants were randomised into (1) DLT with fluoroscopy-guided MLD (intervention group), (2) DLT with traditional MLD (control group), or (3) DLT with placebo MLD (placebo group). Superficial lymphatic architecture was evaluated as a secondary outcome, visualised by ICG lymphofluoroscopy at the baseline (B0), post-intensive (P), and post-maintenance phases (P6). Variables were (1) number of efferent superficial lymphatic vessels leaving the dermal backflow region, (2) total dermal backflow score, and (3) number of superficial lymph nodes. The traditional MLD group showed a significant decrease in the number of efferent superficial lymphatic vessels at P (*p* = 0.026), and of the total dermal backflow score at P6 (*p* = 0.042). The fluoroscopy-guided MLD and placebo group showed significant decreases in the total dermal backflow score at P (*p* < 0.001 and *p* = 0.044, respectively) and at P6 (*p* < 0.001 and *p* = 0.007, respectively); the placebo MLD group showed a significant decrease in the total number of lymph nodes at P (*p* = 0.008). However, there were no significant between-group differences for the changes in these variables. In conclusion, based on lymphatic architecture outcomes, the added value of MLD, in addition to the other parts of DLT, could not be demonstrated in patients with chronic mild to moderate BCRL.

## 1. Introduction

With 2.2 million cases a year worldwide, breast cancer is the most common cancer among women [1]. The survival in general is good, but the treatment has an impact on quality of life. Its medical treatment is associated with several complications such as pain, fatigue [2], and limited shoulder range of motion, leading to a decreased quality of life [3,4]. Around one in five breast cancer survivors develop breast cancer-related lymphoedema (BCRL) at the level of the ipsilateral hand, arm, shoulder, trunk, or any combination of these areas [5]. Lymphoedema is caused by a reduced transport capacity of the lymphatic system (related to surgery, radiotherapy, or both), sometimes combined with an increase in lymph load (for example, related to infection) [6,7].

Near infrared fluorescence imaging of the lymphatic system, also called lymphofluoroscopy, is an imaging technique that can be used to evaluate the superficial lymphatic architecture [8,9]. A tracer, indocyanine green (ICG), is injected intradermally into the patient’s hand. Once excited by a near-infrared light, ICG emits a fluorescent signal and is captured by the camera. In this way, real-time video images of the superficial lymphatic vessels and lymph nodes are provided. Moreover, in patients with lymphoedema, three dysfunctional dermal backflow patterns can be distinguished [10]: (1) splash pattern, representing tortuous dermal dilated lymph capillaries; (2) stardust pattern, which demonstrates spotted fluorescent signals, representing the effusion of lymph fluid out of the lymph capillaries; and (3) diffuse pattern, by which the tracer is widely distributed without identifiable spots and lymph vessels [11]. Lymphofluoroscopy is safe, non-invasive, and has a low cost [9]. A disadvantage is that it only visualises superficial structures and tissue (maximal depth of 2 cm) [8].

**Table 1 cancers-15-01545-t001:** Comparison of the characteristics of fluoroscopy-guided MLD and traditional MLD.

	Fluoroscopy-Guided MLD [12,13]	Traditional MLD [12,14,15]
Stimulation of lymphatic transport in general	Patient-specific, applied on the superficial lymph vessels, lymph nodes and region with dermal backflow, visualized through lymphofluoroscopy	Blind, without knowing the patient-specific lymphatic transport and architecture
Stimulation of resorption by lymph capillaries	Short rolling and stretching with a small surface (e.g., with the thumb) to create a relatively high local pressure	Rolling and stretching movement with the whole hand
Stimulation of transport through lymph collectors	Gliding gently with the medial side of the thumb and lateral side of the index over the lymph collector; light pressure with the hands and limited shear forces on the skin	Pumping with the whole hand (i.e., rolling with the hand from index to little finger) over the lymph collector; light pressure with hands
Stimulation of transport through area with dermal rerouting	Idem stimulation of transport through lymph collectors, but greater pressure with the hands and slower hand movements	Idem stimulation of transport through lymph collectors

MLD = manual lymph drainage.

According to the consensus document of the International Society of Lymphology (ISL), decongestive lymphatic therapy (DLT) is recommended as the treatment for BCRL [16]. It consists of two phases: the intensive phase, which pursues the largest volume reduction, and the maintenance phase, which aims to sustain and optimise the results obtained in the intensive phase. The first phase lasts two to four weeks and consists of skin care, exercise therapy, compression by a multi-component bandage, manual lymph drainage (MLD), and education. The second phase lasts a lifetime and consists of skincare, exercise therapy, compression by a low stretch elastic sleeve, and MLD if needed. There are multiple traditional MLD techniques taught by the Vodder [15], Leduc [17], and Földi [18] MLD schools. A technique that differs from these traditional MLD methods is the Fill and Flush MLD method (developed by Belgrado et al. on a normal lymphatic system), which is also called fluoroscopy-guided MLD [13]. See Table 1 for the difference between fluoroscopy-guided MLD and traditional MLD.

Different systematic reviews show that the added value of traditional MLD in the other parts of DLT for volume reduction is limited [19,20,21,22,23]. Our RCT comparing fluoroscopy-guided MLD with traditional or placebo MLD in addition to DLT was unable to find any difference in volume reduction of the arm or in fluid accumulation at the shoulder/trunk [24]. In contrast, some studies have demonstrated the improvement of the lymphatic transport and architecture after one session of traditional MLD. In two studies, the lymph velocity was increased in healthy volunteers [25,26] and subjects with lymphoedema [26]. Another study showed more visible pathways after one session of traditional MLD [27]. One study showed a positive evolution of the stardust pattern toward a splash pattern after a short period (3 weeks) of daily traditional MLD treatment [28]. However, robust evidence on the long-lasting effect of MLD on the lymphatic architecture has not been provided thus far. 

Therefore, the aim of the present study is to investigate the effectiveness of fluoroscopy-guided MLD versus traditional MLD or placebo MLD, in addition to the other components of DLT, on the superficial lymphatic architecture in breast cancer survivors with chronic mild to moderate BCRL.

We hypothesise that patients who received the fluoroscopy-guided MLD would have a significantly greater improvement in the superficial lymphatic architecture compared to the traditional MLD group or placebo MLD group after three weeks of intensive treatment (P) and after six months of maintenance treatment (P6). This means that patients receiving the fluoroscopy-guided MLD will have (1) a significantly greater increase in the number of efferent superficial lymphatic vessels leaving the dermal backflow region; (2) a significantly greater decrease in the total dermal backflow score; and (3) a significantly greater increase in the number of lymph nodes compared to the traditional MLD group or placebo MLD group.

## 2. Materials and Methods

### 2.1. Study Design and Setting

The Effort-BCRL trial is a multicentre, double-blind randomised controlled trial involving three groups. The details of the trial design are described in detail elsewhere [24]. The results of the primary outcomes and some of the secondary outcomes have already been published [24,29]. The current paper analyses one of the secondary outcomes: the lymphatic architecture visualised through lymphofluoroscopy.

The trial has been approved by the Ethical Committee of the University Hospitals of Leuven (main committee), with subsequent positive advice from the Ethical Committees of each other participating centre (i.e., University Hospital of Saint-Pierre, in General Hospital Groeninge and University Hospital of Ghent, Antwerp University Hospital; CME reference S58689, EudraCT Number 2015-004822-33). The trial has also been registered at clinicaltrials.gov (NCT02609724). The RCT is written according to the CONSORT guideline [30].

### 2.2. Participants

All participants were recruited between February 2016 and September 2019. The inclusion criteria were (1) unilateral lymphoedema of the arm and/or hand, developed after treatment for breast cancer; (2) chronic lymphoedema stage I to IIb (duration of >3 months); (3) at least 5% difference between the arms, adjusted for hand dominance, and/or between the hands; and (4) no active metastases. The exclusion criteria were (1) age < 18 y; (2) oedema of the upper limb from a cause other than breast cancer treatment; (3) mentally or physically unable to participate during the entire study period; (4) allergy to iodine, sodium iodine, or indocyanine green; (5) hyperthyroidism; (6) previous lymph node transplantation or lymphovenous shunt; and (7) bilateral axillary lymph node dissection. Only participants who signed the informed consent form were included in the study.

### 2.3. Intervention

See Figure 1 for an overview of the intervention. All participants received standard DLT consisting of education, skincare, exercises, and compression therapy. The intervention consisted of an intensive phase of 3 weeks and a maintenance phase of 6 months. The participants were randomly assigned to the intervention group receiving fluoroscopy-guided MLD, the control group receiving traditional MLD, or the placebo group receiving placebo MLD.

Treatment sessions were performed by five different physical therapists experienced in the treatment of BCRL. At University Hospitals Leuven, patients were treated by RVH, LB, LV, and AKH; at the University Hospital of Saint-Pierre, by LV and TDV; at the General Hospital Groeninge and University Hospital of Ghent, by LV and TDV; and at the Antwerp University Hospital, by TDV). Prior to the start of the trial, the therapists had multiple training sessions to become familiar with this protocol and ensure identical treatment.

### 2.4. Assessments

#### 2.4.1. Lymphofluoroscopy

To determine the superficial lymphatic architecture, all participants received a standardised lymphofluoroscopy at baseline (B0), both after 3 weeks of intensive treatment (P) and after 6 months of maintenance treatment (P6). For the fluoroscopy-guided MLD group, the baseline lymphofluoroscopy was also used to determine the procedure of MLD (i.e., which hand manoeuvres at which location). Three doctors (ST, LV, and CM) and physical therapists (ND, NG, and KD) performed the lymphofluoroscopic assessments. ST and ND performed the assessments at the University Hospitals Leuven and General Hospital Groeninghe; ST and NG at the Antwerp University Hospital,;CM and ND at the University Hospital of Ghent; and LV and KD at Sint-Pierre University Hospital.

The protocol for the lymphofluoroscopy is shown in Table 2. In summary, a solution with ICG was injected intradermally in the first and fourth web space of the affected hand. The whole procedure consisted of three phases (early phase, break, and late phase). During the early phase, the lymphatic transport was evaluated at rest (3 min), after activity (3 min of flexion/extension of the hand), and after stimulation with MLD (5 min). The second phase consisted of a 60 min break in which exercises (5 min) were alternated with rest (10 min). In the late phase, (1) after making a scan with the camera, the superficial lymphatic architecture was designed on the patient’s arm and trunk and pictures were taken of the ventral, dorsal, and lateral side of the arm and trunk; (2) the lymphatic vessels, lymph nodes, and dermal backflow patterns were designed on a body diagram; and (3) the evaluation sheet was completed.

Based on the study conducted by Thomis et al. on interrater reliability of the scoring of the superficial lymphatic architecture through lymphofluoroscopy [26], the following variables were used:(1)The number of efferent superficial lymphatic vessels leaving the dermal backflow region after the break: The presence and number of efferent superficial lymphatic vessels leaving the dermal backflow region were scored.(2)The total dermal backflow score after the break: This score represents the dermal backflow patterns based on the severity staging of Yamamoto (normal pattern (score 0), splash pattern (score 1), stardust pattern (score 2), and diffuse pattern (score 3)) [27]. The body was divided into thirteen areas, and each area was scored between 0 to 3. The areas were: (1) fingers, (2) dorsal hand, (3) ventral hand, (4) dorsal distal forearm, (5) dorsal proximal forearm, (6) ventral distal forearm, (7) ventral proximal forearm, (8) dorsal distal upper arm, (9) dorsal proximal upper arm, (10) ventral distal upper arm, (11) ventral proximal upper arm, (12) breast region, and (13) scapular region. Per area, the most severe score was withheld; if only a part of an area showed dermal backflow, the dermal backflow pattern score was given for the total area. The total dermal backflow score is the sum of the dermal backflow scores of the thirteen areas of the body (score between 0 and 39). The higher the score, the larger the area with dermal backflow and the more severe the stage of dermal backflow.(3)The number of superficial lymph nodes after the break: The number of retroclavicular, axillar, cubital, and/or humeral lymph nodes were scored. The total number of superficial lymph nodes is the sum of all the different lymph nodes that were visualised.

The data of the variables were extracted from the evaluation sheets and compared with the body diagram. If there was a disagreement between the evaluation sheet and the body diagram, the information on the body diagram was preferred.

#### 2.4.2. Other Assessments

The baseline characteristics were obtained through a clinical evaluation of the participants, including an assessment of body height and weight to determine BMI, an evaluation of the pitting status at the level of the hand, ventral and dorsal lower and upper arm, elbow, shoulder, trunk, and breast (0 = no, 1 = doubtfully present, 2 = clearly present), and this would determine a pitting score between 0 and 18 as well as the lymphoedema stage (ISL stage 1-2b). Moreover, circumferences of the affected and non-affected arms were determined and—based on the formula of the truncated cone—the volumes of the arms were calculated. The volumes of both hands were determined by the water displacement method. By adding the volume of the arm and hand, the total volume of the arm was obtained and the absolute and relative excessive volume of the arm was calculated.

During an interview, the duration of the lymphoedema was questioned. The medical file of the patient was consulted for other relevant information, such as the participant’s age, as well as breast cancer histology, stage, and treatment.

The participants registered in a diary when they performed self-exercises, self-MLD and registered the hours of wearing the compression garment. In this way, the adherence to the self-management was evaluated.

### 2.5. Sample Size Calculation

The study required 201 subjects (67 subjects per group) based on sample size calculation of the primary outcomes, i.e., reduction in excess volume of the arm/hand and accumulation of excess volume at the shoulder/trunk, as well as an alpha of 0.0125 and a power of 80% [24]. This number of subjects was necessary to detect a difference of 15% in the reduction of the excessive volume at the level of the arm/hand or the excessive fluid accumulation at the level of the shoulder/trunk between the three groups. The estimated drop-out rate was 5% based on a previous longitudinal study with breast cancer patients [28]. There was no sample size calculation performed for the outcome of this paper, as it demonstrates the secondary outcome of the EFforT-BCRL trial.

### 2.6. Randomization and Allocation Sequence Generation

All participants were assigned to one of the three groups. The allocation was randomly computer-generated by using 6-size permuted blocks, and then performed by an independent person (ADG). The randomisation was determined by an identification number, which was given to the participants after inclusion in the study.

### 2.7. Blinding

The assessors were blinded for the allocation of the participants to the intervention groups. In addition, the participants were blinded successfully during their allocation to the intervention groups. At the end of the follow-up, only 41 of 180 participants (23%) could indicate their allocated group correctly [19].

### 2.8. Statistical Analyses

Analyses of the number of efferent superficial lymphatic vessels leaving the dermal backflow region, the total dermal backflow score, and the total number of lymph nodes were based on raw scores. A multivariate linear model for longitudinal measures was used to compare the evolution of the raw scores between the three groups (fluoroscopy-guided MLD vs. traditional MLD vs. placebo MLD). An unstructured covariance matrix was used for the 3 × 3 covariance matrix of the repeated measures over time (B0, P, P6) for the three variables. At each time point, changes versus the baseline were calculated and compared between the three groups. *p*-values for the overall interaction (group × time) effect were measured. Given that a likelihood procedure was used, subjects with incomplete outcome information were also included in the analysis. Alpha levels were set at 5%. There were no corrections performed for multiple testing of the secondary outcomes.

All analyses were performed by using IBM SPSS Statistics software, version 27, for Windows.

## 3. Results

### 3.1. Participants

Of the 391 patients that were screened, 194 were included in the present study (UH Leuven, *n* = 112; UH Saint-Pierre, *n* = 10; UH Antwerp, *n* = 35; UH Ghent, *n* = 14; and GH Groeninge, *n* = 23). Among these, 65 patients were randomised to the intervention group, 64 patients to the control group, and 65 patients to the placebo group. Of all 194 included patients, 5 (of the 9 that were estimated) participants dropped out during the intensive treatment phase. Of these, 4 were lost to follow-up. Figure 2 shows the overall flow of the subjects in the trial up to 6 months of maintenance treatment. Of all participants, 65 were randomised to the intervention group, 64 to the control group, and 65 to the placebo group. Only 4 and 7 participants did not receive a lymphofluoroscopy after the intensive phase (P) and 6 months of maintenance (P6), respectively. None of the participants reported an adverse event caused by the lymphofluoroscopy. The characteristics of all participants and for each group separately are given in Table 3.

### 3.2. Change of Superficial Lymphatic Architecture

Table 4 demonstrates the change in the lymphatic architecture over time using a comparison between the intervention group, control group, and placebo group.

#### 3.2.1. Number of Efferent Superficial Lymphatic Vessels Leaving Dermal Backflow Region

In the whole group, the number of efferent lymphatic vessels did not significantly change between the baseline and the post-intensive phase, nor between baseline and the 6 months of maintenance time interval (*p* > 0.05). The change in the number of efferent lymphatic vessels from the baseline to the post-intensive phase and 6 months of maintenance was not significantly different between the 3 groups (*p* = 0.406). However, in the group with traditional MLD (control group), a significant decrease in the number of efferent lymphatic vessels was observed between the baseline and the post-intensive phase (*p* = 0.026), as well as a borderline significant decrease between the baseline and 6 months of maintenance (*p* = 0.089).

The hypothesis that patients receiving the fluoroscopy-guided MLD (intervention group) had a significantly greater increase in the number of efferent superficial lymphatic vessels leaving the dermal backflow region compared to patients receiving traditional MLD (control group) or placebo MLD (placebo group) could not be withheld.

#### 3.2.2. Total Dermal Backflow Score

In the whole group, the dermal backflow score decreased significantly between the baseline and post-intensive phase (*p* = 0.001), as well as between baseline and 6 months of maintenance (*p* < 0.001). This decrease in the dermal backflow score was also found in the three groups separately. However, the change in the total dermal backflow score over time was not significantly different between the three groups (*p* = 0.268).

The hypothesis that patients receiving the fluoroscopy-guided MLD (intervention group) had a significantly greater decrease in the total dermal backflow score than patients receiving traditional MLD (control group) or placebo MLD (placebo group) could not be withheld.

#### 3.2.3. Total Number of Lymph Nodes

In the whole group, the total number of lymph nodes decreased significantly from the baseline to the post-intensive timepoint (*p* = 0.008). The change in the total number of lymph nodes was not significantly different between the three groups either (*p* = 0.642). However, in the placebo group, there was a significant decrease in the number of lymph nodes between the baseline and the post-intensive phase (from 0.7 to 0.4; *p* = 0.008).

The hypothesis that patients receiving fluoroscopy-guided MLD (intervention group) had a significantly greater increase of the number of lymph nodes than patients receiving the traditional MLD (control group) or placebo MLD (placebo group) could not be withheld.

## 4. Discussion

This is the first randomised controlled trial with a large sample size investigating the short- and long-term added value of MLD to the other parts of DLT for superficial lymphatic architecture.

The present study was unable to demonstrate any added value of fluoroscopy-guided MLD compared to traditional MLD or placebo MLD for the superficial lymphatic architecture in the short term (post-intensive phase) or in the long term (after 6 months of maintenance). Consequently, the three hypotheses were rejected. There was a significant decrease in the dermal backflow score in the three groups. However, this change was not significantly different between the three groups. Similarly to our study, Medina-Rodríguez et al. performed intensive DLT consisting of MLD, pneumatic compression, multi-layer bandaging, and exercises over the course of 3 weeks. They also found an improvement of the dermal backflow pattern [23]. This is the only study apart from ours that has investigated the effect of different sessions of DLT on changes in the lymphatic architecture. However, they did not include a control group, as all participants received MLD. As a consequence, they were unable to state that the improvement of the dermal backflow pattern was due to MLD. Moreover, follow-up was limited to 3 weeks, and the number of patients was small (*n* = 19). All other studies investigated the effect of only one session of MLD on the lymphatic transport/architecture [20,21,22,29]. In the present study, no significant between-group differences for the change of the number of efferent superficial lymphatic vessels, nor for the number of superficial lymph nodes over time, was found. On the contrary, a decrease was observed in the number of lymph vessels outside of the dermal backflow region for the traditional MLD group (control group), as was a decrease in the number of superficial lymph nodes for the placebo group, which may indicate deterioration of the lymphatic transport. The results of these within-group differences must be interpret with caution, because there were no corrections performed for multiple testing.

### 4.1. Strengths and Limitations

The study has several strengths. First, the risk of performance bias was minimal. More than 75% of the patients were blinded for the allocation to the treatment group [19]. Second, the medical doctors/physical therapists performing the lymphofluoroscopy were blinded for the treatment allocation as well. Third, interrater reliability of the scoring of the lymphatic architecture by ICG lymphofluoroscopy was proven by the investigator team [26]. Fourth, to guarantee standardization of the treatments, they were performed by a group of physical therapists who were trained before the start of the study. Fifth, the drop-out rate was very low. Sixth, patients were taught how to perform self-MLD (according to their method, i.e., fluoroscopy-guided MLD, traditional MLD, or placebo MLD) during the maintenance phase, when there was no treatment given by a therapist. In this way the study obtains the maximum benefit from the MLD treatment. Finally, in comparison with other randomised controlled trials, this study has a large sample size and a follow-up period of 6 months. A limitation of this study is that the calculation of the sample size was based on the primary outcomes of the EFforT-BCRL trial [24], not on the secondary endpoint reported in this trial. Therefore, the significance of results in this paper should be interpreted with caution.

### 4.2. Clinical Implications

Analyses of the primary outcomes of the EFforT-BCRL trial showed no additional effect of MLD (neither fluoroscopy-guided nor traditional) on arm volume reduction and changes in lymph stagnation at the level of the shoulder and trunk, in addition to the other components of DLT [19]. These results are in line with the results of previous systematic reviews [14,15,16,17,18]. Analyses of our secondary outcomes showed no additional effects of MLD on the quality of life, problems with functioning, local tissue water, extracellular fluid, skin thickness, or skin elasticity at the arm or trunk level in the short or long term [30]. In line with the primary analyses and other secondary analyses, the current paper was unable to demonstrate an added value of MLD for superficial lymphatic architecture. This means that in patients with chronic BCRL, MLD does not provide a clinically relevant additional benefit nor a physiological benefit when added to other components of DLT. As a consequence, there is no evidence to include (fluoroscopy-guided or traditional) MLD in the treatment of patients with chronic BCRL.

## 5. Conclusions

This trial was not able to demonstrate any beneficial value of MLD for the other components of DLT in terms of outcomes regarding the superficial lymphatic architecture of patients with chronic, mild to moderate BCRL.

## Figures and Tables

**Figure 1 cancers-15-01545-f001:**
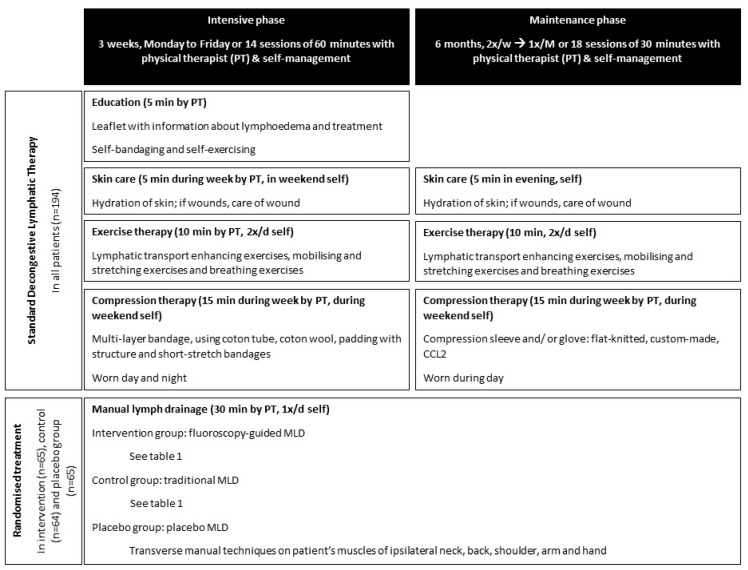
Overview of the intervention in the trial (PT = physical therapist; CCL = compression class; MLD = manual lymph drainage).

**Figure 2 cancers-15-01545-f002:**
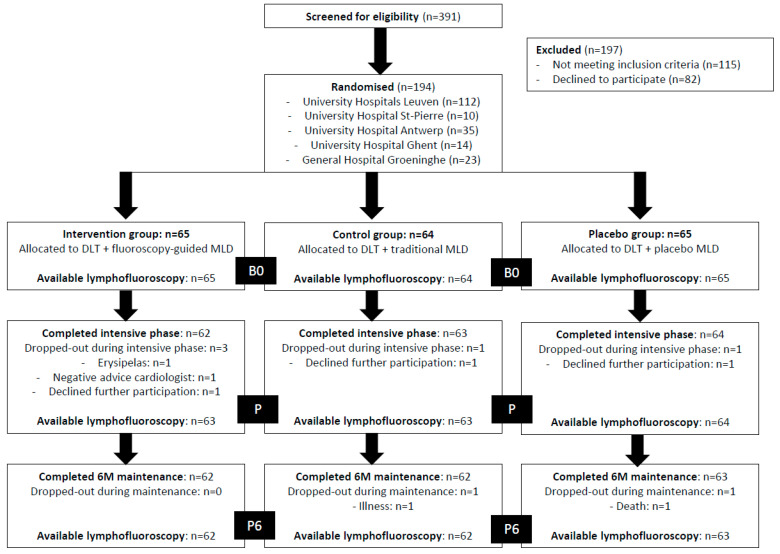
Overview of the study flow (DLT = decongestive lymphatic therapy; MLD = manual lymph drainage; B0 = baseline; P = post-intensive phase; P6 = after 6 months of maintenance phase).

**Table 2 cancers-15-01545-t002:** Protocol of lymphofluoroscopy and the procedure to determine the different variables.

	Step	Duration	Description
Preparation	0.1 Dilution of ICG		Suspended ICG in 25 mL pure water and subsequently diluted with saline water to reach a final concentration of ICG of 0.08 mg/mL per injection site
	0.2 Camera		Participant in sitting position with arms on table; camera is held perpendicular to the observed skin at a distance of 15 cm (best focus)
	0.3 Injection of ICG		Intradermal injection of 0.2 mL ICG solution in 1st (ulnar injection point) and 4th web space (radial injection point) dorsally in the hand
Early phase	1.1 Rest	3 min	Hand in resting position
	1.2 Activity	3 min	Participant performs flexion/extension of the hand, with a large range of motion and lower arm stable on table
	1.3 Stimulation	5 min	Therapist performs manual techniques on the skin: fills lymph capillaries at the level of the injection points and stimulates transport through the lymph collectors and dermal backflow
	1.4 Drawing on body diagram and completing evaluation sheet	5 min	Is not used in the present paper
Break	2.1 Pressure and activity	1 h	- Therapist places a piece of foam on the injection points; an elastic bandage (Mollelast, Lohmann, and Rauscher) is placed around the hand to increase the pressure on the injection points; - Participant performs exercises: alternately, 5 min of squeezing hand, 10 min of rest, 5 min of circumduction with hand, 10 min rest, etc.
Late phase	3.1 Scan with camera	20 s	- Dorsal arm (with hand in pronation) up to retroclavicular region; - Ventral arm (with hand in supination) up to axilla (at the end lift of arm); - Scapular region: from ipsilateral axilla to vertebral column; - Pectoral region: from ipsilateral axilla to sternum;
	3.2 Drawing on skin and body diagram	10 min	- Medical doctor/therapist draws, under fluorescence feedback, the lymph collectors and regions with dermal backflow on the participant’s skin; pictures are taken of the ventral, dorsal, and lateral side of the arm and trunk - Lymph vessels and regions of dermal backflow (splash, stardust, and diffuse) are designed on a body diagram
	3.3 Completing evaluation sheet	5 min	- Number of lymph vessels (out of injection points and out of dermal backflow) - Presence of splash, stardust, and diffuse dermal backflow patterns and location (fingers, hand, proximal/distal and ventral/dorsal lower or upper arm, breast, and trunk) - Number of lymph nodes (cubital, humeral, axillary, retroclavicular)

ICG = indocyanine green.

**Table 3 cancers-15-01545-t003:** Characteristics of the participants.

Variable	Total Group (*n* = 194)	Intervention Group (*n* = 65) DLT + Fluoroscopy Guided MLD Group	Control Group (*n* = 64) DLT + Traditional MLD	Placebo Group (*n* = 65) DLT + Placebo MLD
	(mean (±SD) or median [IQR] *)	(mean (±SD) or median [IQR] *)	(mean (±SD) or median [IQR] *)	(mean (±SD) or median [IQR] *)
Body Mass Index (kg/m^2^)	28.1 (±5.7)	27.6 (±5.3)	28.8 (±5.6)	27.8 (±6.1)
Age (years)	61.1 (±9.8)	60.3 (±10.8)	61.8 (±9.5)	61.1 (±9.0)
Duration of lymphoedema (months) *	24 [58]	29 [49]	28 [73]	16 [50]
Absolute excessive arm volume (mL) *	441.0 [442.3]	456.7 [390.5]	441.8 [464.4]	430.0 [510.8]
Relative excessive arm volume (%) *	21.7 [19.9]	22.8 [24.2]	21.9 [20.5]	21.0 [18.9]
Total pitting score (/18) at baseline *	5 [5]	5 [4]	5 [5]	4 [6]
	*n (%)*	*n (%)*	*n (%)*	*n (%)*
Site enrolment				
University Hospitals Leuven	112 (57.7%)	39 (60%)	36 (56.3%)	37 (56.9%)
University Hospital Antwerp	35 (18%)	9 (13.8%)	10 (15.6%)	16 (24.6%)
University Hospital St-Pierre Brussels	10 (5.2%)	6 (9.2%)	2 (3.1%)	2 (3.1%)
General Hospital Groeninge Kortrijk	23 (11.9%)	7 (10.8%)	7 (10.9%)	7 (10.8%)
University Hospital Ghent	14 (7.2%)	4 (6.2%)	9 (14.1%)	3 (4.6%)
Gender				
Male	1 (0.5%)	0 (0.0%)	1 (1.6%)	0 (0.0%)
Female	193 (99.5%)	65 (100.0%)	63 (98.4%)	65 (100.0%)
Lymphoedema stage				
Stage I	32 (16.5%)	10 (15.4%)	10 (15.6%)	12 (18.5%)
Stage IIa	109 (56.2%)	34 (52.3%)	40 (62.5%)	35 (53.8%)
Stage IIb	53 (27.3%)	21 (32.3%)	14 (21.9%)	18 (27.7%)
Type of breast surgery				
Mastectomy	115 (59.3%)	36 (55.4%)	40 (62.5%)	39 (60%)
Breast-conserving surgery	79 (40.7%)	29 (44.6%)	24 (37.5%)	26 (40%)
Tumour stage				
pT1	58 (29.9%)	20 (30.7%)	20 (31.3%)	17 (26.2%)
pT2	104 (53.6%)	32 (49.2%)	29 (45.3%)	43 (66.2%)
pT3	18 (9.3%)	6 (9.2%)	9 (14.1%)	3 (4.6%)
pT4	14 (7.2%)	7 (10.8%)	6 (9.3%)	2 (3.1%)
Lymph node stage				
pN0	45 (23.2%)	12 (18.5%)	16 (25%)	15 (23.1%)
pN1	99 (51.5%)	36 (55.4%)	32 (50%)	34 (52.3%)
pN2	26 (13.4%)	11 (16.9%)	8 (12.5%)	7 (10.8%)
pN3	23 (11.9%)	6 (9.2%)	8 (12.5%)	9 (13.8%)
Metastasis	3 (1.5%)	1 (1.5%)	0 (0.0%)	2 (3.1%)
Adjuvant treatment				
Radiotherapy	189 (97.4%)	63 (96.9%)	63 (98.4%)	63 (96.9%)
Chemotherapy	167 (86.1%)	57 (83.1%)	52 (81.2%)	61 (93.8%)
Endocrine therapy	152 (78.4%)	51 (78.5%)	53 (82.8%)	48 (73.8%)
Target therapy	39 (20.1%)	13 (20.0%)	12 (18.8%)	14 (21.5%)

DLT = decongestive lymphatic therapy; MLD = manual lymph drainage; SD = standard deviation; IQR = interquartile range; T = tumour stage; N = lymph node stage; * Not normal distributed variables, which are reported as median [IQR].

**Table 4 cancers-15-01545-t004:** Change of the lymphatic architecture from baseline to post-intensive phase and 6 months post-maintenance, with a comparison between the 3 groups.

	Total Group (*n* = 194)	Intervention Group (*n* = 65) DLT + Fluoroscopy Guided MLD Group	Control Group (*n* = 64) DLT + Traditional MLD	Placebo Group (*n* = 65) DLT + Placebo MLD	P overall Interaction
	Estimate (CI)	Estimate (CI)	Estimate (CI)	Estimate (CI)	
Number of lymphatic vessels leaving dermal backflow region					
Baseline (B0)	1.3 (1.0; 1.5)	1.1 (0.7; 1.5)	1.6 (1.2; 2.0)	1.0 (0.6; 1.4)	0.406
After intensive phase (P)	1.1 (0.9; 1.3)	1.0 (0.8; 1.3)	1.2 * (0.8; 1.5)	1.1 (0.7; 1.4)	
After 6M maintenance (P6)	1.1 (0.9; 1.4)	1.0 (0.6; 1.4)	1.3 (0.9; 1.6)	1.2 (0.8; 1.5)	
Dermal backflow score (0–51)					
Baseline (B0)	9.5 (8.9; 10.1)	10.2 (9.1; 11.3)	9.0 (7.8; 10.0)	9.3 (8.3; 10.4)	0.268
After intensive phase (P)	8.8 * (8.2; 9.4)	9.0 ** (8.0; 9.9)	8.9 (7.9; 9.9)	8.6 * (7.6; 9.5)	
After 6M maintenance (P6)	8.2 ** (7.7; 8.8)	8.5 ** (7.5; 9.4)	8.1 * (7.1; 9.0)	8.1 * (7.2; 9.1)	
Number of lymph nodes					
Baseline (B0)	0.6 (0.4; 0.7)	0.7 (0.1; 0.6)	0.7 (0.5; 1.0)	0.7 (0.4; 0.9)	0.642
After intensive phase (P)	0.4 * (0.3; 0.6)	0.3 (0.1; 0.5)	0.6 (0.4; 0.8)	0.4 * (0.2; 0.6)	
After 6M maintenance (P6)	0.5 (0.4; 0.6)	0.3 (0.1; 0.6)	0.6 (0.4; 0.8)	0.518 (0.3; 0.8)	

DLT = decongestive lymphatic therapy; MLD = manual lymph drainage. Estimate = estimated marginal means; CI = 95% confidence interval. * Statically significant changes (*p* < 0.05) in estimated mean within groups versus baseline. ** Statically significant changes (*p* < 0.001) in estimated mean within groups versus baseline.

## Data Availability

Data supporting the reported results can be made available upon request.

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
