# Peer review of "The Effectiveness of Fluoroscopy-Guided Manual Lymph Drainage as Part of Decongestive Lymphatic Therapy on the Superficial Lymphatic Architecture in Patients with Breast Cancer-Related Lymphoedema: A Randomised Controlled Trial"

_cancers, 2023, doi:10.3390/cancers15051545_

Round 1
Reviewer 1 Report
I congratulate the authors for their work in this study, a very well structured trial, with a well-based objective and a well designed methodology.
Nevertheless, I would like to give some feedback about certain issues that should be addressed before considering this manuscript for publication.
Keywords:
1. The terms “lymphatic architecture”, “near infrared fluorescence lymphatic imaging” and “lymphofluoroscopy” could not be found in the Mesh database, so I suggest to change them for indexed terms.
Introduction:
2. In my opinion there are very few references for the paragragh between lines 54 and 62. Please, add more references, there are plenty of studies related to the epidemiology of lymphoedema.
3. Table 1, please refer the resource.
Materials and methods:
4. You refer as numer 24 the details of the trial, but I am not able to find it anywhere. Please, provide that complementary information.
5. You refer the CONSORT guidelines as numer 25, but references end at number 23.
6. The recruiting phase ended in 2019, three years ago. Why haven´t these results been published before? Please, justify the reason why these results are being published so long after the experiment was over.
7. In line 139, there is an extra “.” Before the word Intervention.
8. There are no complementary files or data about references number 26-28.
Results:
9. Previously, in Methods, the authors said the trial sample had to be 201, and in the size-sample calcultation, I understood they had added a 5% to counter drop-out. But in results, the final sample was 194, and I can´t see in the flowchart the reasons of drop-out to understand why there are less than 201 participants. I can only see that a certain number of participants either didn´t meet the criteria, or had refused to participate in the trial, that´s all. So, from the very beginning of the study, were there less participants than 201?
Author Response
Thank you so much for your positive comment.
Keywords:
- The terms “lymphatic architecture”, “near infrared fluorescence lymphatic imaging” and “lymphofluoroscopy” could not be found in the Mesh database, so I suggest to change them for indexed terms.
That is indeed a good suggestion. I have changed the keywords in the paper on p. 2 in ‘lymphatic system’ (instead of lymphatic architecture) and added ‘molecular imaging’, ‘indocyanine green’. I have kept manual lymph drainage and near infrared fluorescence lymphatic imaging, as these are paper-specific terms for which no MesH terms exist.
Introduction:
- In my opinion there are very few references for the paragragh between lines 54 and 62. Please, add more references, there are plenty of studies related to the epidemiology of lymphoedema.
This is correct. We have tried to reduce the amount of references to keep us to the most recent published reviews and meta-analyses in these areas. However, to accommodate your request, we added more references.
Lovelace, D.L.; McDaniel, L.R.; Golden, D. Long-Term Effects of Breast Cancer Surgery, Treatment, and Survivor Care. J Midwifery Womens Health 2019, 64, 713-724, doi:10.1111/jmwh.13012.
Verbelen, H.; Gebruers, N.; Eeckhout, F.M.; Verlinden, K.; Tjalma, W. Shoulder and arm morbidity in sentinel node-negative breast cancer patients: a systematic review. Breast Cancer Res Treat 2014, 144, 21-31, doi:10.1007/s10549-014-2846-5.
Grada, A.A.; Phillips, T.J. Lymphedema: Pathophysiology and clinical manifestations. J Am Acad Dermatol 2017, 77, 1009-1020, doi:10.1016/j.jaad.2017.03.022.
- Table 1, please refer the resource.
We have made up this table ourselves. We have added references in the table.
For the explanation of fluoroscopy-guided MLD we refer to the protocol of the paper and to the paper of Belgrado et al.
For explanation of traditional MLD we refer to the protocol of the Effort-BCRL trial and the paper of Kasseroller and book of Lee (where Leduc explains MLD).
Materials and methods:
- You refer as number 24 the details of the trial, but I am not able to find it anywhere. Please, provide that complementary information.
De Vrieze T, Gebruers N, Nevelsteen I, et al. Manual lymphatic drainage with or without fluoroscopy guidance did not substantially improve the effect of decongestive lymphatic therapy in people with breast cancer-related lymphoedema (EFforT-BCRL trial): a multicentre randomised trial. J Physiother 2022;68(2):110-122, doi:10.1016/j.jphys.2022.03.010
De Vrieze T, Gebruers N, Nevelsteen I, et al. Does Manual Lymphatic Drainage Add Value in Reducing Arm Volume in Patients With Breast Cancer-Related Lymphedema? Phys Ther 2022, doi:10.1093/ptj/pzac137
- You refer the CONSORT guidelines as number 25, but references end at number 23.
This is indeed a mistake. I have added the correct reference.
- The recruiting phase ended in 2019, three years ago. Why haven´t these results been published before? Please, justify the reason why these results are being published so long after the experiment was over.
The reason why this paper was not sooner being prepared for publication, is that we first focused on writing the publication regarding the primary outcomes (published in Journal of Physiotherapy) and a paper regarding all but one secondary outcomes (published in Physical Therapy). We opted to write a separate paper regarding this particular secondary outcome (i.e. lymphatic architecture), due to the originality of the topic and the extensiveness of data.
- In line 139, there is an extra “.” Before the word Intervention.
Thanks for mentioning, this has been deleted.
- There are no complementary files or data about references number 26-28.
I do not understand this comment. We have referred to these papers in the introduction.
Results:
- Previously, in Methods, the authors said the trial sample had to be 201, and in the size-sample calcultation, I understood they had added a 5% to counter drop-out. But in results, the final sample was 194, and I can´t see in the flowchart the reasons of drop-out to understand why there are less than 201 participants. I can only see that a certain number of participants either didn´t meet the criteria, or had refused to participate in the trial, that´s all. So, from the very beginning of the study, were there less participants than 201?
Yes, this is correct. As depicted in the flowchart, 194 patients in total were included in the trial and were measured during a baseline assessment. This means that the inclusion of patients was ended before the project’s predefined number of patients (n=201) was reached. Possibly due the relatively strict inclusion and exclusion criteria, the accrual rate was slower than anticipated. Sixty-four subjects were needed in each group to detect a difference of 15 percentage points in the relative reduction of the excessive volume (see protocol paper). Therefore, the planned sample size was increased to 67 subjects per group to anticipate potential drop-out. Nevertheless, although the study was terminated earlier (i.e. after inclusion of 194 subjects), this did not jeopardize the power of the primary analysis, since this analysis was still based on information from 194 subjects at baseline (65, 64, 65 in the 3 groups, respectively) and 190 subjects (63, 63, 64 in the 3 groups, respectively) after 3 weeks of intensive treatment
We have changed the text as follow:
Of the 391 patients that were screened, 194 were included in the present study (UH Leuven; n=112, UH Saint-Pierre; n=10, UH Antwerp; n=35, UH Ghent; n=14 and GH Groeninge; n=23). Among these, 65 patients were randomised to the intervention group, 64 patients to the control group and 65 patients to the placebo group. Of all 194 included patients, 5 (of the 9 that were estimated) participants dropped-out during the intensive treatment phase. Of them, 4 were lost to follow-up.The flow of participants during the trial is presented in Figure 2.

Reviewer 2 Report
This study is a well conducted and interesting article. I think it might be welcome in the context of breast cancer research. However, I have some recommendations, as follows:
1. Perhaps the title needs to be modified to be more succinct, for example “The Effectiveness of Fluoroscopy-Guided Manual Lymph Drainage as Parts of The Decongestive Lymphatic Therapy on The Superficial Lymphatic Architecture in Patients with Breast Cancer-Related Lymphoedema: A Randomised Controlled Trial”
2. Added the full names of the abbreviations for Figure 1-2, and Table 1-3 in the right place such as figure captions, footnote of tables as it presented in Table 4.
3. It seems that there is ambiguity in the sentence Line 273-Line 276 “However, in the group with traditional MLD (control group) a significant decrease in the number of efferent lymphatic vessels was observed between baseline to the post-intensive phase (p=0.026) and to 6 months of maintenance (p>0.050).”, please rephrase it.
4. There exists some minor format errors, such as Line 128 (2.2..), Line 139 (2.3..) , Line 155 (2.4..), and Line 156 (2.4.1..)
5. The manuscript needs to be revised in order to correct English language mistakes. For example:
(Lin54) Breast cancer is with 2.2 million cases a year worldwide the most common cancer among women: rephrase sentence
6. Moreover, I’m not sure whether the baseline characteristics of the participants between the three groups are balanced or not, and I think different adjuvant therapy might have potential inference on change of superficial lymphatic architecture when we take comparison between the intervention group, control group and placebo group.
Author Response
Thank you for your suggestions and for reading our paper.
- Perhaps the title needs to be modified to be more succinct, for example “The Effectiveness of Fluoroscopy-Guided Manual Lymph Drainage as Parts of The Decongestive Lymphatic Therapy on The Superficial Lymphatic Architecture in Patients with Breast Cancer-Related Lymphoedema: A Randomised Controlled Trial”
Good suggestions. We will shorten the title.
- Added the full names of the abbreviations for Figure 1-2, and Table 1-3 in the right place such as figure captions, footnote of tables as it presented in Table 4.
Done!
- It seems that there is ambiguity in the sentence Line 273-Line 276 “However, in the group with traditional MLD (control group) a significant decrease in the number of efferent lymphatic vessels was observed between baseline to the post-intensive phase (p=0.026) and to 6 months of maintenance (p>0.050).”, please rephrase it.
There was only a significant decrease in number of efferent lymphatic vessels between baseline and the post-intensive phase (p=0.026) and a borderline significant decrease between baseline and 6 months of maintenance (p=0.089).
- There exists some minor format errors, such as Line 128 (2.2..), Line 139 (2.3..) , Line 155 (2.4..), and Line 156 (2.4.1..)
Many thanks for these remarks. The additional ‘.’ In these lines have been deleted.
5. The manuscript needs to be revised in order to correct English language mistakes. For example: (Lin54) Breast cancer is with 2.2 million cases a year worldwide the most common cancer among women: rephrase sentence
Done. We have changed it in: With 2.2 million cases a year worldwide, breast cancer is shown to be the most common cancer among women
This sentence has been rephrased.
- Moreover, I’m not sure whether the baseline characteristics of the participants between the three groups are balanced or not, and I think different adjuvant therapy might have potential inference on change of superficial lymphatic architecture when we take comparison between the intervention group, control group and placebo group.
As presented in Table 3 (Characteristics of participants), the numbers regarding the adjuvant therapy are perfectly balanced between the three groups. According to our statistician, it is not necessary to compare the baseline data statistically, but one has to consider the clinically relevant differences. In our opinion, there are no clinically relevant differences at baseline.

Reviewer 3 Report
- Titles of each subsection in the Results section should be substitute by a sentence stating the results obtained.
- A concluding sentence is needed at the end of each subsection in the Results section.
- The Introduction and the Discussion provide sufficient information to understand the state-of-the-art and citations are appropiate. Limitations are highlighted.
- This article will benefit from some experimental validation (qPCR, flow cytometry, Western Blot, any kind of staining...)
Author Response
- Titles of each subsection in the Results section should be substitute by a sentence stating the results obtained.
- A concluding sentence is needed at the end of each subsection in the Results section
We do not understand this comment. Do you suggest to remove the different headings? If this is what you mean, they can be removed as in the text the content of the heading is repeated. There is already a conclusion at the end of every section (this is the section whether the hypothesis is confirmed/ rejected).
- The Introductionand the Discussion provide sufficient information to understand the state-of-the-art and citations are appropriate. Limitations are highlighted.
Thank you.
- This article will benefit from some experimental validation (qPCR, flow cytometry, Western Blot, any kind of staining...)
Is this a comment related to our paper? Our statistician advised us regarding the kind of statistical analyses and never mentioned this kind of analyses.
